# Direct and indirect salt effects on homotypic phase separation

**Matt MacAinsh[1], Souvik Dey[1], Huan-Xiang Zhou[1,2]***

[1]Department of Chemistry, University of Illinois Chicago, Chicago, United States;
[2]Department of Physics, University of Illinois Chicago, Chicago, United States

## eLife Assessment

In this potentially **important** study, the authors conducted atomistic simulations to probe the salt-dependent phase separation of the low-complexity domain of hnRN-PA1 (A1-LCD). The authors have identified both direct and indirect mechanisms of salt modulation, provided explanations for four distinct classes of salt dependence, and proposed a model for predicting protein properties from amino acid composition. There is a range of opinions regarding the strength of evidence, with some considering the evidence as **incomplete** due to the limitations in the length and statistical errors of the computationally intense atomistic MD simulations.

***For correspondence:**
hzhou43@uic.edu

**Abstract** The low-complexity domain of hnRNPA1 (A1-LCD) phase separates in a salt-dependent manner. Unlike many intrinsically disordered proteins (IDPs) whose phase separation is suppressed by increasing salt concentrations, the phase separation of A1-LCD is promoted by >100 mM NaCl. To investigate the atypical salt effect on A1-LCD phase separation, we carried out all-atom molecular dynamics simulations of systems comprising multiple A1-LCD chains at NaCl concentrations from 50 to 1000 mM NaCl. The ions occupy first shell as well as more distant sites around the IDP chains, with Arg sidechains and backbone carbonyls the favored partners of $Cl^-$ and $Na^+$, respectively. They play two direct roles in driving A1-LCD condensation. The first is to neutralize the high net charge of the protein (+9) by an excess of bound $Cl^-$ over $Na^+$; the second is to bridge between A1-LCD chains, thereby fortifying the intermolecular interaction networks in the dense phase. At high concentrations, NaCl also indirectly strengthens $\pi$–$\pi$, cation–$\pi$, and amino–$\pi$ interactions, by drawing water away from the interaction partners. Therefore, at low salt, A1-LCD is prevented from phase separation by net charge repulsion; at intermediate concentrations, NaCl neutralizes enough of the net charge while also bridging IDP chains to drive phase separation. This drive becomes even stronger at high salt due to strengthened $\pi$-type interactions. Based on this understanding, four classes of salt dependence of IDP phase separation can be predicted from amino-acid composition.

## Introduction

Biomolecular condensates formed via liquid–liquid phase separation (LLPS) mediate a variety of cellular functions such as biogenesis of the ribosome and stress response (*Mitrea et al., 2018*; *Protter and Parker, 2016*). The driver for phase separation or condensation is intermolecular interactions, including electrostatic, hydrogen bonding, $\pi$–$\pi$, cation–$\pi$, amino–$\pi$, and hydrophobic (*Das et al., 2020*; *Dignon et al., 2020*; *Zhou et al., 2024*). Salt can tune all these interactions and thus exert significant effects on phase separation. While the screening effect of salt on electrostatic interactions is well-known, its effects on other types of interactions may be indirect and perhaps are less appreciated. In particular, high salt strengthens hydrophobic interactions by increasing the surface tension of water (*Baldwin, 1996*; *Zhou and Pang, 2018*). Because salt can exert disparate effects on different

types of interactions, it can be used as a perturbation to dissect the relative importance of these interactions in phase separation (*Farag et al., 2023*; *Hazra and Levy, 2023*). Inside cells, proteins can encounter varying salt conditions at different locations or at different times, and therefore the drive for their phase separation can span a wide range.

Numerous studies of salt effects on protein phase separation have been reported. The most typical effect is the suppression of phase separation by screening electrostatic attraction, both for homotypic systems (*Berry et al., 2015*; *Zhang et al., 2015*; *Brady et al., 2017*; *Strom et al., 2017*; *Wei et al., 2017*; *Reed and Hammer, 2018*; *Tsang et al., 2019*; *Martin et al., 2021*; *Lin et al., 2024*) and for heterotypic systems (*Farag et al., 2023*; *Galvanetto et al., 2023*). However, salt can also promote phase separation, although the mechanism is not always clear (*Martin et al., 2021*; *Muschol and Rosenberger, 1997*; *Burke et al., 2015*; *Kim et al., 2017*; *Dao et al., 2018*; *Babinchak et al., 2019*; *Le Ferrand et al., 2019*; *Wong et al., 2020*; *Agarwal et al., 2021*; *Krainer et al., 2021*; *Otis and Sharpe, 2022*). In particular, *Krainer et al., 2021* observed a 'reentrant' salt effect on the phase separation of five intrinsically disordered proteins (IDPs): phase separation occurs without salt, is prohibited by medium salt, and reemerges at high salt. They attributed the reemergence of phase separation to strengthened π-type and hydrophobic interactions that overcompensate weakened electrostatic attraction. A unifying understanding of how salt affects the phase separation of IDPs is still lacking. For example, it is an open question whether salt effects on phase separation can be predicted from the protein sequence. Deep knowledge, in particular at the atomic level, of how salt affects intermolecular interactions and, ultimately, phase separation is required.

The low-complexity domain of hnRNPA1 (A1-LCD) represents another IDP where an atypical salt effect was reported (*Martin et al., 2021*). The full-length protein, comprising folded domains along with the LCD, phase separates in the absence of salt and the tendency to phase separate is reduced upon adding salt, thus exhibiting the typical salt effect. A screening mechanism, specifically of electrostatic attraction between the LCD and the folded domains, is supported by small-angle X-ray scattering and coarse-grained molecular dynamics (MD) simulations. In contrast, A1-LCD does not phase separate without salt and starts to do so only after 100 mM NaCl is added. An earlier study revealed that π-types of interactions, mediated by aromatic residues, drive the phase separation of A1-LCD (*Martin et al., 2020*). A follow-up study, based on charge mutations, further showed that the net charge plays a strong suppressive role (*Bremer et al., 2022*). The saturation concentration ($C_{sat}$) for phase separation is minimum at a net charge near 0, and increases by two orders of magnitude, signifying an enormous weakening of the drive for phase separation, when the net charge moves away from neutrality in either direction. In a more recent study, salt promoted the homotypic phase separation of both A1-LCD and FUS-LCD, but suppressed the heterotypic phase separation of their mixture (*Farag et al., 2023*). The salt effect on the phase separation of A1-LCD was modeled by a Debye–Hückel potential in coarse-grained simulations (*Tesei and Lindorff-Larsen, 2022*). Similarly, the salt effect on the phase separation of another IDP, Ddx4, was analyzed using the random phase approximation based on a coarse-grained representation (*Lin et al., 2024*; *Lin et al., 2016*; *Lin et al., 2020*). Coarse-grained simulations with explicit water and ions have been used to study salt effects in both homotypic and heterotypic phase separation (*Garaizar and Espinosa, 2021*).

All-atom MD simulations can uniquely provide mechanistic insight into the driving force and properties of biomolecular condensates (*Zhou et al., 2024*). For example, these simulations showed that ATP, a small molecule with a −4 charge, bridges between positively charged IDP chains in driving phase separation (*Kota et al., 2024*). The intermolecular interactions quickly break and reform, explaining why the condensates can rapidly fuse despite very high macroscopic viscosity. Similarly, quick breakup and reformation of salt bridges in a heterotypic condensate allow the protein molecules to be extremely dynamic in a highly viscous environment (*Galvanetto et al., 2023*). Recently, all-atom MD simulations provided explanations for wide variations in phase equilibrium and material properties among condensates of tetrapeptides with different amino-acid compositions (*Zhang et al., 2024*). These and other simulations (*Rekhi et al., 2024*) show that all attractive residue–residue contacts contribute to the drive for phase separation.

Here, we study salt effects on A1-LCD condensation by all-atom MD simulations. The simulations reveal two direct effects and one indirect effect of NaCl: neutralization of net charge and bridging between protein chains and strengthening of π-type interactions by drawing water away from the

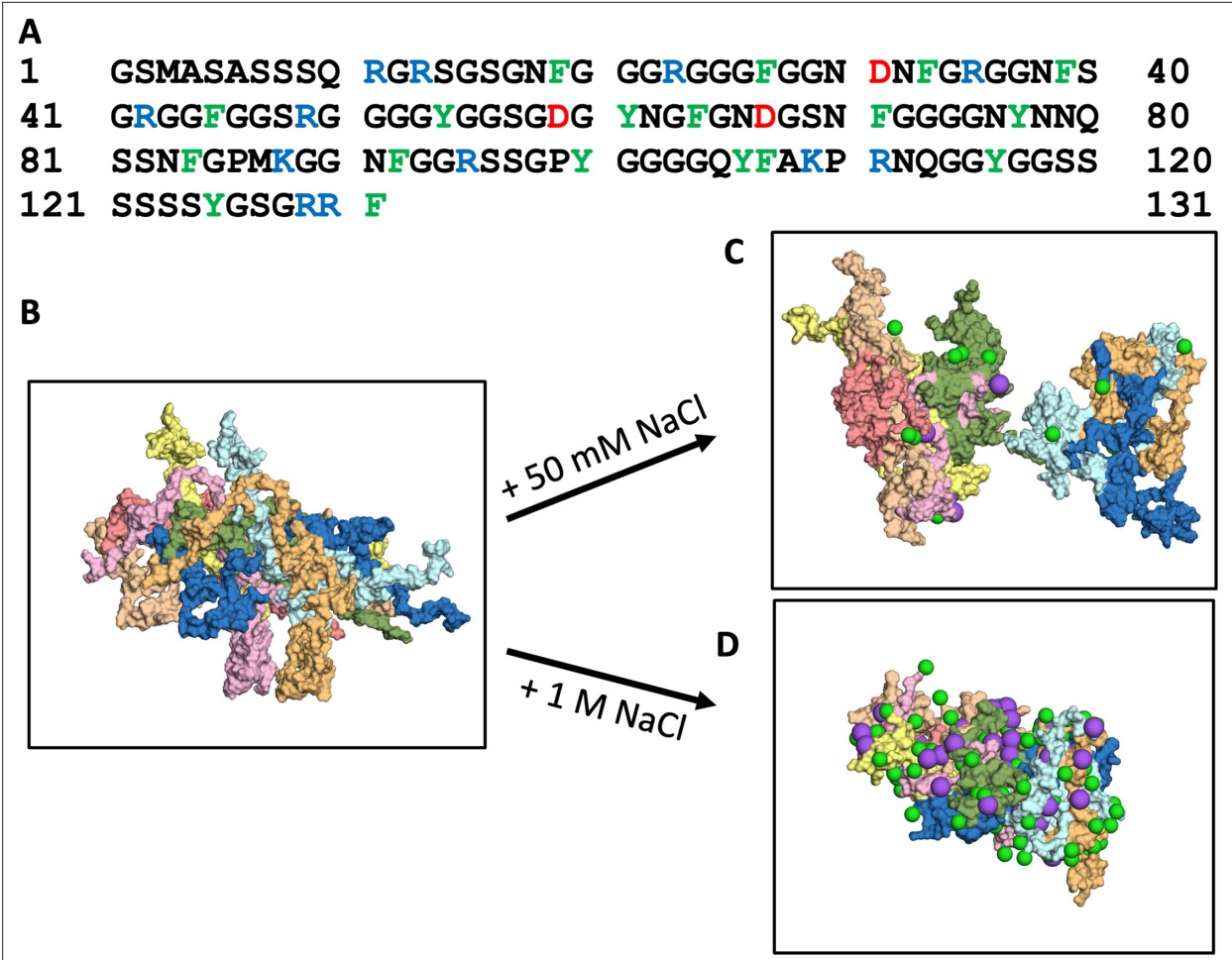

**Figure 1.** Amino-acid sequence of A1-LCD and molecular dynamics simulations of its condensation. (**A**) Amino-acid sequence. (**B**) First frame and (**C, D**) a frame at 1000 ns from 1.5-µs simulations of the 8-chain systems at low and high salt. In all figures, Cl⁻ or Na⁺ ions are represented by green and magenta spheres, respectively.

The online version of this article includes the following source data and figure supplement(s) for figure 1:

**Figure supplement 1.** Salt dependence of the average root-mean-square-fluctuation (RMSF) among the eight chains.

**Figure supplement 1—source data 1.** Source data for *Figure 1—figure supplement 1*.

interaction partners. We also present a unified picture of salt dependences of phase separation by defining four distinct classes and predict these classes from amino-acid composition.

## Results

### Salt condenses A1-LCD and increases inter-chain interactions

The 131-residue A1-LCD is comprised mostly of Gly and Ser (51 and 22, respectively), followed by 18 aromatic residues (11 Phe and 7 Tyr), 17 residues with sidechain amides (13 Asn and 4 Gln), and 15 charged residues (10 Arg, 2 Lys, and 3 Asp), with a large net charge of +9 (*Figure 1A*). Using initial conformations of A1-LCD from the previous single-copy simulations (*Dey et al., 2022*), we built an 8-copy model for the dense phase (with an initial concentration of 3.5 mM) at five NaCl concentrations ranging from 50 to 1000 mM (*Figure 1B*). This initial concentration is close to the measured concentration at the critical point (*Martin et al., 2020*), thereby facilitating the comparison between low salt (where phase separation does not occur) and high salt (where phase separation does occur). Each system was simulated in four replicates for 1.5 µs each. All the results reported here are averages over the four replicate simulations. From the same initial configuration with very few inter-chain

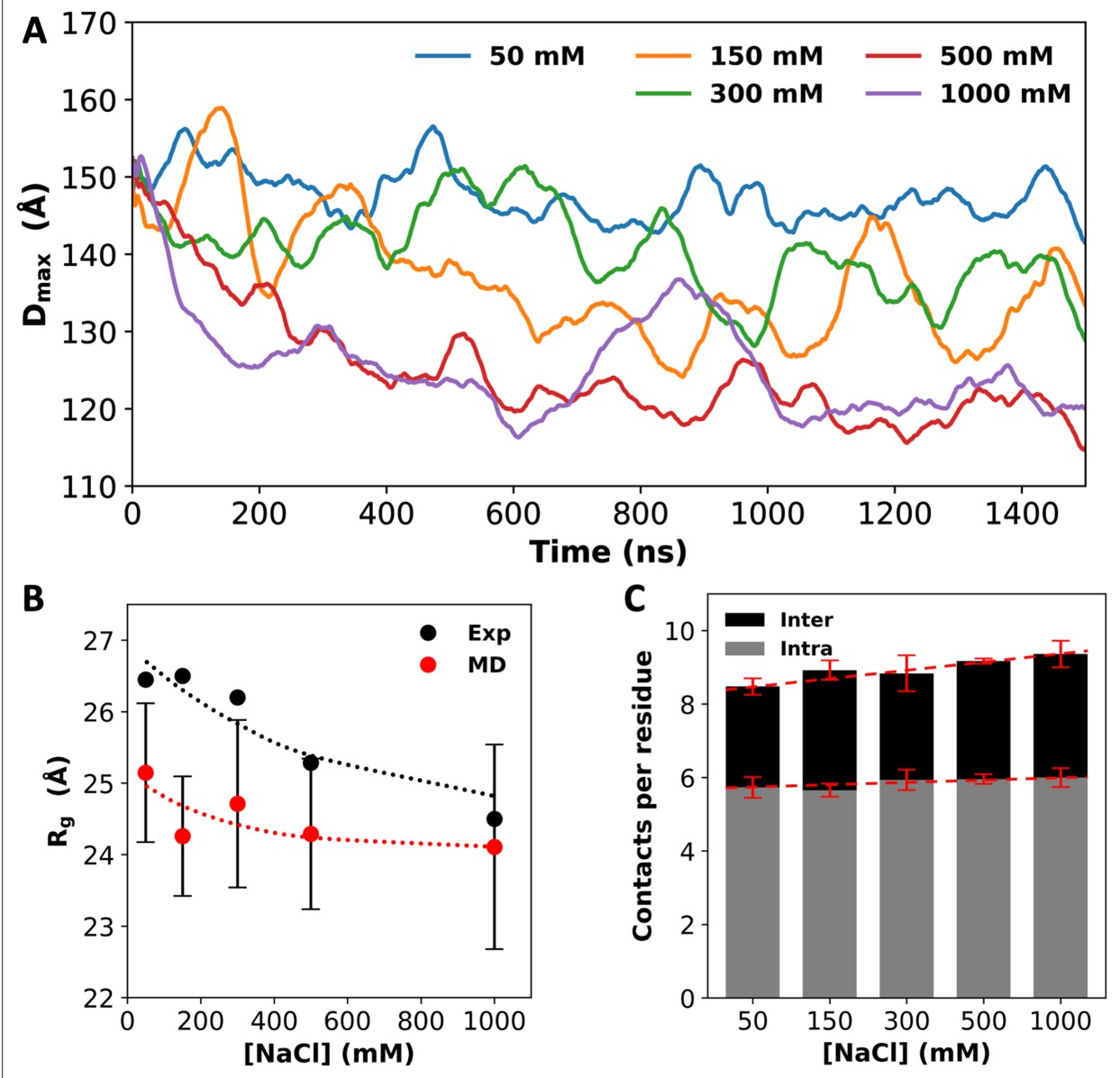

**Figure 2.** Salt effects on A1-LCD condensation and inter-chain interactions. (**A**) $D_{max}$ values, averaged over four replicates, as a function of simulation time. (**B**) Radii of gyration ($R_g$) from small-angle X-ray scattering (**Martin et al., 2021**) and from molecular dynamics (MD) simulations. Dotted curves are drawn to guide the eye. (**C**) Average number of inter- or intrachain contacts per residue at each salt concentration. Dashed lines are drawn to show trends. Error bars represent standard deviations among four replicate simulations.

The online version of this article includes the following source data for figure 2:

**Source data 1.** Source data for **Figure 2**.

contacts, the systems maintain the initial loose configuration at 50 mM NaCl (**Figure 1C**) but condense noticeably at 1000 mM NaCl (**Figure 1D**). At low salt, the protein chains have a tendency to fill the simulation box, occasionally spanning the entire box in one or two of the three orthogonal directions and exhibiting larger-scale reconfiguration based on root-mean-square-fluctuation (RMSF) calculation (**Figure 1—figure supplement 1**). In contrast, at high salt, the chains aggregate into an ellipsoidal particle, with the RMSF reduced by 27%.

We also quantified the contrast between low and high salt by calculating $D_{max}$, the maximum side length of the rectangular box that parallels the simulation box and circumscribes the multi-chain system. The $D_{max}$ values are close to the initial value (~150 Å) at low salt (50 mM NaCl), but decrease to ~135 Å at intermediate salt (150 and 300 mM NaCl) and further to ~120 Å at high salt (500 and

1000 mM NaCl; *Figure 2A*). Overall, the systems show a lack of condensation at low salt, and growing condensation on going to intermediate and high salt. Modeling the condensed particle as an ellipsoid with the principal diameters given by the maximum dimensions in the three orthogonal directions, we can estimate its concentration to be 23 mM, which is similar to the measured concentration in the dense phase (*Martin et al., 2020*). A small part of the reason for the growing condensation is the compaction of the individual chains. The average radius of gyration ($R_g$) shows a decreasing trend with increasing salt (*Figure 2B*), which matches the experimental data (*Martin et al., 2021*). Correspondingly, the average number of intrachain contacts per residue increases slightly, by 5%, when the salt concentration is increased from 50 to 1000 mM (*Figure 2C*). However, the main driver of the condensation is inter-chain interactions, with inter-chain contacts per residue increasing by 23% over the same salt range. Clearly, high salt induces A1-LCD condensation, with an increased number of interactions between protein chains. Below we present the molecular mechanisms for these effects.

## Ions selectively bind to backbone and sidechain sites to neutralize the net charge of A1-LCD

The large net charge of A1-LCD implies significant electrostatic repulsion between chains. Potentially salt ions can neutralize the net charge. To investigate ion–protein binding, we calculated radial distribution functions (RDFs) of $Cl^-$ and $Na^+$ around polar groups of A1-LCD. At 1000 mM NaCl, the RDFs of $Cl^-$ show a strong 1st peak at 3.2 Å around Arg and Lys sidechain nitrogens and a moderate 1st peak around the Gln and Asn sidechain nitrogens and the Ser sidechain oxygen (*Figure 3—figure supplement 1A*). Each Arg sidechain often coordinates two $Cl^-$ ions simultaneously, but each Lys sidechain coordinates only one $Cl^-$ ion. A 2nd peak, at ~5 Å, is also observed around Arg and Lys sidechain nitrogens. Of the remaining polar groups, only the $Cl^-$ RDFs around the Tyr sidechain oxygen and the backbone nitrogen show a weak 1st peak (*Figure 3—figure supplement 1B*). We used cutoffs of 4 and 6.4 Å, respectively, to define 1st- and 2nd-shell $Cl^-$ binding. On a per-residue basis, Arg and Lys sidechains coordinate the most 1st-shell $Cl^-$ ions, reaching ~0.25 ions per residue (*Figure 3A*, left panel). In comparison, each Gln, Asn, or Ser residue, on average, coordinates ~0.05 $Cl^-$ ions. Given the large numbers of Arg and Ser residues in the A1-LCD sequence, these two residue types coordinate the most 1st-shell $Cl^-$ ions, 21.5 and 8.4, respectively, in the 8-chain system at 1000 mM NaCl. Although the RDF around backbone nitrogens has only a weak 1st peak, given their large number (131 per chain), they actually coordinate 13.4, or the second most 1st-shell $Cl^-$ ions. Among these 13.4 $Cl^-$ ions, 55% are coordinated with Gly residues. A large part of this high percentage is due to the enrichment of Gly (39% of all residues) in the A1-LCD sequence, but Gly is additionally favored for its lack of sidechain, which allows the close approach of ions to the backbone.

$Na^+$ shows a very strong preference for the Asp carboxyl in 1st-shell coordination, with a peak RDF value of 9.2 at 2.2 Å (*Figure 3—figure supplement 2A*). Each Asp sidechain carboxyl typically coordinates a single $Na^+$ ion, usually in a bifurcated geometry. In addition, $Na^+$ shows a strong preference in 1st-shell coordination with Asn and Gln sidechain amide oxygens and the backbone carbonyl oxygen as well as a moderate preference with the Ser sidechain oxygen. A 2nd peak, at ~4.2 Å, is also observed in the $Na^+$ RDF around the Asp carboxyl. Of the remaining polar groups, only a weak 1st peak is seen in the RDF around the Tyr sidechain oxygen (*Figure 3—figure supplement 2B*). We used cutoffs of 3 and 5.4 Å, respectively, to define 1st- and 2nd-shell $Na^+$ binding. On a per-residue basis, the Asp sidechain coordinates the most 1st-shell $Na^+$ ions, reaching 0.17 ions per residue (*Figure 3A*, right panel). The per-residue $Na^+$ ion number reduces to ~0.06 for Asn and Gln sidechain oxygens and further to ~0.03 for the backbone oxygen and Ser sidechain oxygen. Again, due to their large number, backbone oxygens coordinate a very large number, 27.9, of 1st-shell $Na^+$ in the 8-chain system at 1000 mM NaCl. This number dwarfs the counterparts for sidechains, the largest of which are 7.4, 4.1, and 4.3 respectively, for Asn, Asp, and Ser. For coordination with the backbone, similar to the case with $Cl^-$, 55% of the 27.9 $Na^+$ ions involve Gly residues, again reflecting the fact that this amino acid allows the close approach of ions to the backbone. Overall, the largest number of 1st-shell $Cl^-$ ions are coordinated with Arg sidechains but the largest number of 1st-shell $Na^+$ ions are coordinated with backbone carbonyls. Whereas each Arg sidechain often coordinates two $Cl^-$ ions, multiple backbone carbonyls often coordinate a single $Na^+$ ion, with at least two backbone carbonyl partners for 30.6 of the 64.1 bound $Na^+$ ions (1st- and 2nd-shell).

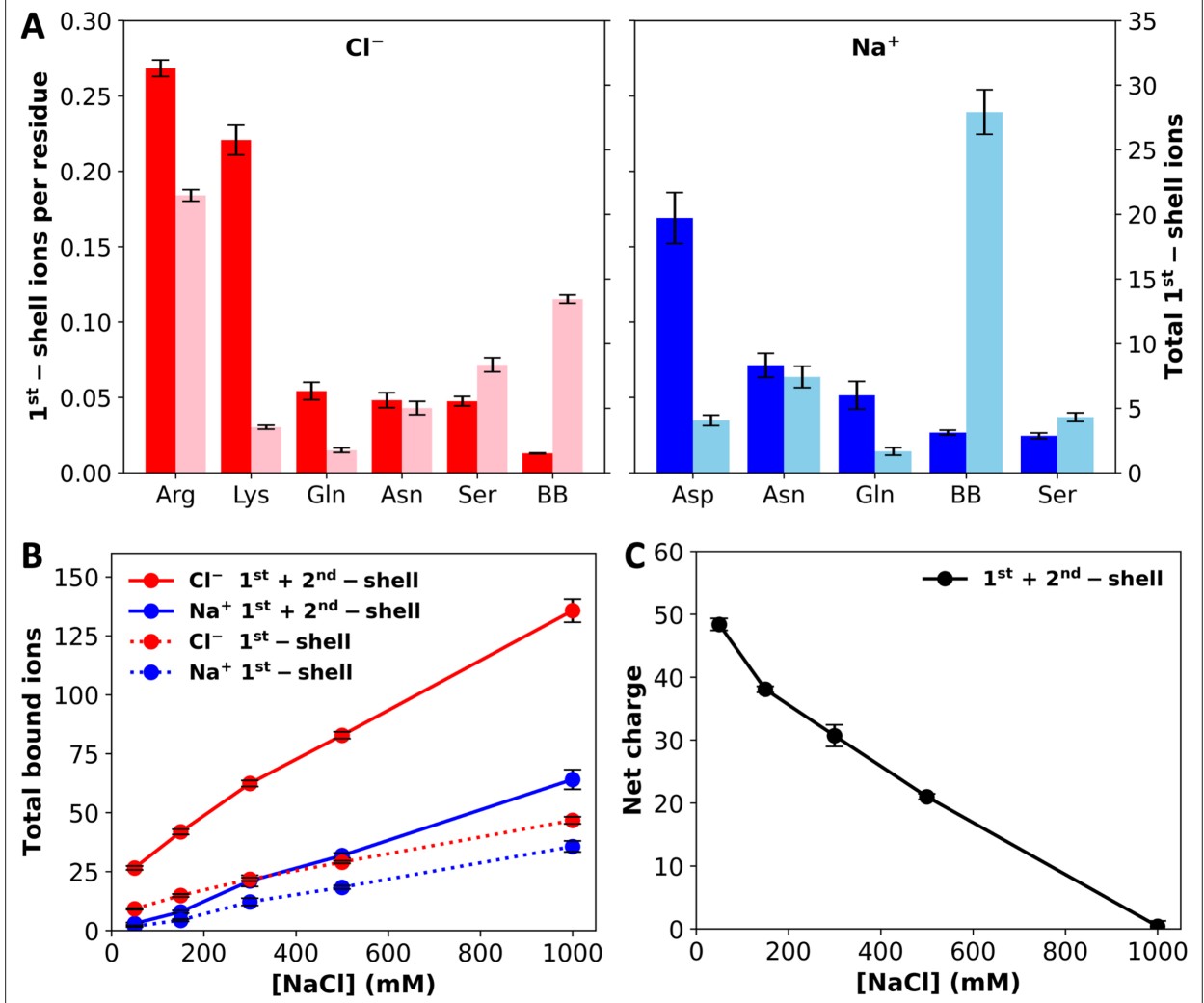

**Figure 3.** Levels of ion binding at different NaCl concentrations. (**A**) Number of 1st-shell ions, with (darker colors; tick marks on the left vertical axis) and without (lighter colors; tick marks on the right vertical axis) normalization by the number of residues of a given amino-acid type, at 1000 mM NaCl. (**B**) Total number of 1st-shell only or 1st- and 2nd-shell ions. (**C**) Net charge of the system with 1st- and 2nd-shell ions included. Error bars represent standard deviations among four replicate simulations.

The online version of this article includes the following source data and figure supplement(s) for figure 3:

**Source data 1.** Source data for *Figure 3*.

**Figure supplement 1.** Radial distributions functions of $Cl^-$ around sidechain and backbone N and O atoms.

**Figure supplement 1—source data 1.** Source data for *Figure 3—figure supplement 1*.

**Figure supplement 2.** Radial distributions functions of $Na^+$ around sidechain and backbone O and N atoms.

**Figure supplement 2—source data 1.** Source data for *Figure 3—figure supplement 2*.

In *Figure 3B*, we display the total numbers of 1st-shell $Cl^-$ and $Na^+$ ions in the 8-chain systems as a function of NaCl concentration. Both the 1st-shell $Cl^-$ and $Na^+$ ions increase with increasing NaCl concentration, and $Cl^-$ ions always outnumber $Na^+$ ions. The difference remains almost constant, with around 10 more 1st-shell $Cl^-$ ions than 1st-shell $Na^+$ ions. This difference is not enough to neutralize the total charge, +72, on the protein chains. However, when 2nd-shell ions are also included, the excess of bound $Cl^-$ ions over bound $Na^+$ ions grows quickly with increasing NaCl concentration. Correspondingly, the net charge of the system reduces to zero at 1000 mM NaCl (*Figure 3C*). Net charge repulsion explains why the chains maintain their initial loose configuration in the simulations at 50 mM NaCl (*Figure 1C*) and the observed absence of phase separation at this low salt concentration (*Martin et al., 2021*). At high salt, the protein net charge is completely neutralized by 1st- and

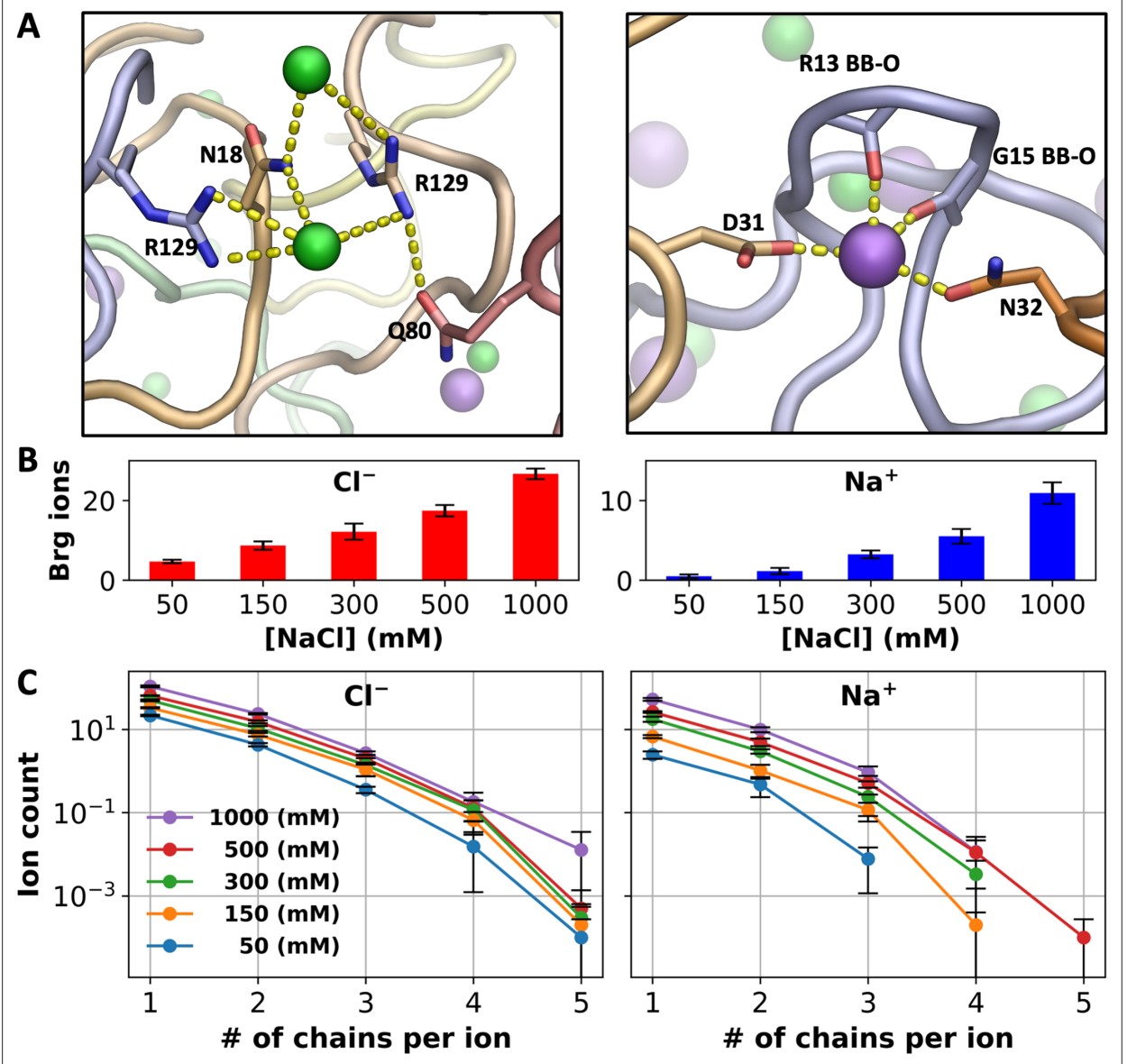

**Figure 4.** Bridging ions. (**A**) Examples of chain bridging by $Cl^-$ and $Na^+$; $Cl^-$ ions are coordinated by Arg and other sidechains whereas $Na^+$ ions are coordinated by both backbone carbonyls (including from Gly) and sidechain oxygens. (**B**) Average number of $Cl^-$ or $Na^+$ ions engaged in bridging between A1-LCD chains. (**C**) Average number of ions bound in 1st- and 2nd-shell sites lined by a given number of A1-LCD chains. Error bars represent standard deviations among four replicate simulations.

The online version of this article includes the following source data for figure 4:

**Source data 1.** Source data for *Figure 4*.

2nd-shell ions; hence the protein chains condense (*Figure 1D*) and phase separation was readily observed (*Martin et al., 2021*).

## Ions act as bridges between protein chains to drive condensation

In addition to charge neutralization, we suspected that ions could also fortify intermolecular interactions by bridging between protein chains, similar to the role played by ATP molecules in driving phase separation of positively charged IDPs (*Kota et al., 2024*). Indeed, we found that $Cl^-$ has a tendency to bind with Arg and other sidechains from multiple chains and likewise, $Na^+$ has a tendency to bind with backbone carbonyls and sidechain oxygens from multiple chains (*Figure 4A*).

To quantify this tendency, we calculated the number of bridging $Cl^-$ or $Na^+$ ions, i.e., those that bind (both 1st- and 2nd-shell) to residues on more than one A1-LCD chain (*Figure 4B*). Close to 20% of all 1st- and 2nd-shell ions bridge between A1-LCD chains. For both $Cl^-$ and $Na^+$, the average number of bridging ions increases with increasing salt, but the pace of increase is much greater for $Na^+$ than for $Cl^-$, commensurate with the trend shown by the total number of bound ions of each type (*Figure 3B*). The number of bridging ions is 4.7 for $Cl^-$ and only 0.5 for $Na^+$ in the 8-chain system at 50 mM NaCl, and increases to 26.6 and 10.9, respectively, at 1000 mM NaCl. The greater fold change of bridging $Na^+$ ions (23-fold, vs 6-fold for $Cl^-$) is also apparent when we break the bridging ions according to the number of A1-LCD chains being bridged (*Figure 4C*). For $Cl^-$, the curves plotting the ion count against the number of bridged chains are close to each other among the different NaCl concentrations, but the counterparts for $Na^+$ are more spread out.

These disparate effects of salt concentration on chain bridging by $Cl^-$ and $Na^+$ can be explained by the difference in coordination properties between the two ion types presented above. $Cl^-$ strongly prefers Arg sidechains and, even at low salt, occupies a large number of sites lined by them, of which ~20% are bridging sites. However, there is a relatively limited supply of Arg sidechains (a total of 80 in the 8-chain system) and hence the fold change in bridging $Cl^-$ ions upon salt increase is somewhat tempered. In contrast, although $Na^+$ prefers Asp sidechains, there are very few of those in the system; instead, $Na^+$ predominantly binds to backbone carbonyls. As the latter binding is relatively weak, the number of bridging $Na^+$ ions is very small at low salt. However, since there is a large supply of backbone carbonyls (a total of 1048 in the 8-chain system), at high salt, $Na^+$ ions bind to a portion of these backbone carbonyls and bridge A1-LCD chains. Of the total 10.9 $Na^+$ bridging ions, 6.4 do so through backbone carbonyls of at least two chains. Consequently, at low salt, chain bridging is dominated by $Cl^-$ ($Na^+$ only 9% of bridging ions), but at high salt, $Na^+$ becomes more even (30% of bridging ions) with $Cl^-$ in bridging A1-LCD chains. An important reason for the latter is $Na^+$ coordination by backbone carbonyls, especially those of Gly residues (*Figure 4A*).

## Salt also contributes to condensation indirectly by strengthening π-type interactions

As noted above, the number of inter-chain interactions increases by 23% when the NaCl concentration increases from 50 mM to 1000 mM. While this result could be accounted for by the two direct salt effects presented so far, that is, charge neutralization and chain bridging, which act to condense A1-LCD, there are additional factors. A breakdown of inter-chain sidechain interactions into different types reveals that when the salt concentration is increased from 50 to 1000 mM, the number of salt bridges per chain remains nearly constant, while the numbers of cation-π, π-π, and amino-π interactions increase by 17%, 26%, and 39%, respectively (*Figure 5A*). That is, as A1-LCD is condensed at high salt, there is an overall increase in inter-chain interactions, but this increase is limited to π-types of interactions and excludes salt bridges. At increasing salt, more π-types of interactions are formed while no new salt bridges are formed, suggesting a strengthening of the former interactions.

The apparent null effect of salt on salt bridge formation can be attributed to the canceling of two opposing effects: chain condensation by the direct effects of salt may potentially shorten the distances between salt-bridge partners and thereby strengthen salt bridges but competition of ions with the salt-bridge partners for coordination (*Figure 3—figure supplements 1A and 2A*) may potentially weaken salt bridges. Indeed, at 1000 mM NaCl, cationic and anionic partners in a salt bridge are often found to also coordinate $Cl^-$ and $Na^+$, respectively (*Figure 5B*). In comparison, one partner in cation–π interactions and both partners in π–π and amino–π interactions have limited abilities to coordinate with ions, so the effects from ion competition are tempered.

For π-types of interactions, instead of a weakening mechanism through ion competition, salt may exert a strengthening effect by drawing water away from the interaction partners (*Figure 5D*). We calculated the RDFs of water around Tyr sidechains that form π-types of interactions and found decreases in water density when the salt concentration is increased from 50 to 1000 mM (*Figure 5E* and *Figure 5—figure supplement 1A–C*). The decreases in water density follow the order cation–π < π–π < amino–π, which is the same order as the corresponding increases in these three types of interactions. As a control, no decrease in water density is seen around Asp sidechains that form salt bridges (*Figure 5—figure supplement 1D*). Water can interfere with and weaken π-types of interactions; by drawing water away from the interaction partners, high salt strengthens these interactions

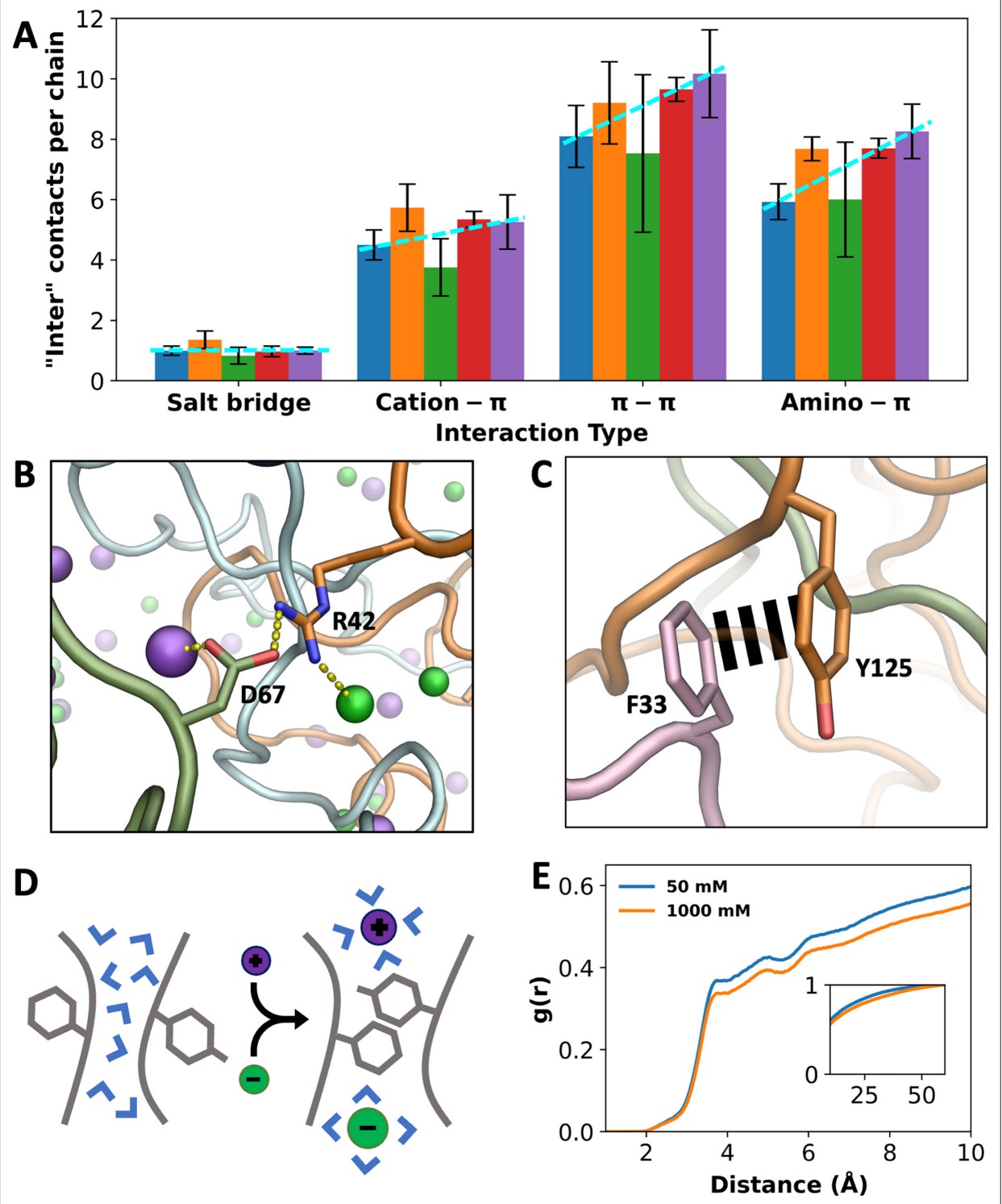

**Figure 5.** Indirect effects of ions on different types of interactions. (**A**) Number of inter-chain contacts per chain for each interaction type. Bars from left to right correspond to increasing salt concentrations (50, 150, 300, 500, and 1000 mM). Dashed lines are drawn to show trends. Error bars represent standard deviations among four replicate simulations. (**B**) An inter-chain salt bridge, with ion coordination by the partner sidechains. (**C**) An inter-chain π–π interaction, free of ion coordination. (**D**) Schematic showing a π–π interaction facilitated by high salt, via drawing water away from the interaction partners. (**E**) Radial distribution functions of water around Tyr residues that interact with Phe, Arg, Lys, Gln, and Asn. Lower values at high salt demonstrate water withdrawal. Inset shows radial distribution functions approaching 1.

The online version of this article includes the following source data and figure supplement(s) for figure 5:

*Figure 5 continued on next page*

*Figure 5 continued*

**Source data 1.** Source data for *Figure 5*.

**Figure supplement 1.** Radial distribution functions (RDFs) of water around sidechains that form interactions with other sidechains.

**Figure supplement 1—source data 1.** Source data for *Figure 5—figure supplement 1*.

and thereby indirectly contributes to condensation. Recently the withdrawal of water from π-types of interaction partners in FUS-LCD condensates has been directly demonstrated by Raman spectroscopy (*Joshi et al., 2024*).

## Discussion

We have shown in atomistic detail the actions of ions in the condensation of A1-LCD over a wide range of NaCl concentrations. The MD simulations reveal that NaCl has both direct and indirect effects in driving phase separation. The first direct effect is to neutralize the net charge of A1-LCD and thereby attenuate net charge repulsion. The second direct effect is to bridge between protein chains and thereby fortify intermolecular interaction networks. In addition, high salt strengthens π-types of interactions by drawing water away from the interaction partners, thereby also indirectly driving phase separation. The net result is that, while phase separation of A1-LCD is prevented by net charge repulsion at low salt, it is enabled at intermediate salt through charge neutralization and chain bridging by ions. The drive for phase separation becomes even stronger at high salt, where π-types of interactions are strengthened.

These findings broaden our understanding of the roles of charges and ions in phase separation. That high net charge is a suppressive factor is highlighted by the strong effect of A1-LCD charge mutations on $C_{sat}$ (*Bremer et al., 2022*). Here, we have shown that this suppressive action can be countered by the addition of salt, which exerts both direct and indirect effects. One of these direct effects, that is, neutralization of net charge, can be treated by a Debye–Hückel potential in coarse-grained simulations (*Tesei and Lindorff-Larsen, 2022*). However, our all-atom explicit-solvent simulations have revealed not only an additional direct effect, that is, bridging between protein chains, but also an indirect effect, that is, strengthening of π-type interactions. In lattice Monte Carlo simulations of a system of charged homopolymer chains plus counterions with Coulomb interactions between all charges, counterions were found to occupy sites between chains and produce effective chain–chain attraction

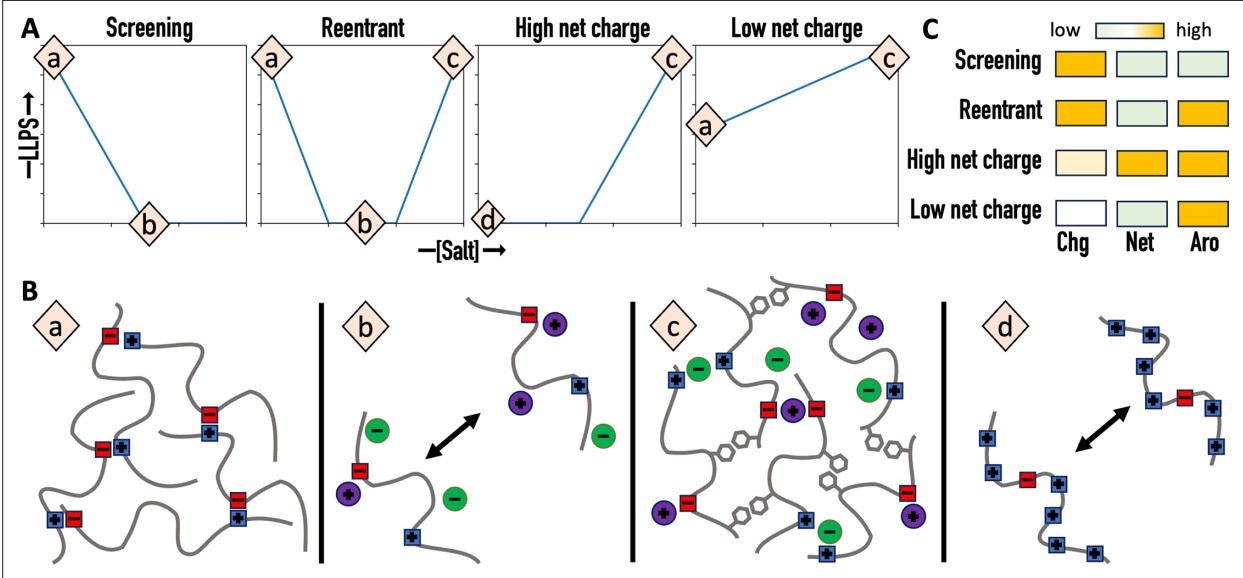

**Figure 6.** Four classes of salt dependence and their prediction from amino-acid composition. (**A**) Salt dependences of liquid–liquid phase separation (LLPS). (**B**) Charge–charge and π-type interactions and their regulation by salt. (**a**) Significant charge–charge attraction. (**b**) Screening of charge-charge attraction by salt. (**c**) Strengthening of π-type interactions by high salt. (**d**) Repulsion due to high net charge. (**C**) Distinctions of the four classes of salt dependence by three determinants: charged content (Chg), net charge (Net), and aromatic content (Aro).

(*Orkoulas et al., 2003*); this effect might be viewed as a primitive form of bridging. *Krainer et al., 2021* observed a reentrant salt effect on the phase separation of IDPs, and attributed the reemergence of phase separation at high salt to strengthened π-type and hydrophobic interactions that overcompensate weakened electrostatic attraction. Their conclusion was based on the salt dependences of the potentials of mean force calculated for pairs of sidechains of various types. Instead of a single pair of sidechains, our simulations are on multiple copies of protein chains. We show that, at high salt, the number of salt bridges remains constant while the numbers of π-types of interactions are elevated in the multi-chain system. Our simulations further reveal that high salt achieves the latter effect by drawing water away from π-interaction partners.

The present work puts us in a position to paint a unified picture of salt dependences of homotypic phase separation (*Figure 6A, B* and *Table 1*). The salt dependence that is most often reported is screening, where salts weaken the electrostatic attraction between protein chains and thereby suppress phase separation (*Berry et al., 2015*; *Zhang et al., 2015*; *Brady et al., 2017*; *Strom et al., 2017*; *Wei et al., 2017*; *Reed and Hammer, 2018*; *Tsang et al., 2019*; *Martin et al., 2021*; *Lin et al., 2024*). The reentrant salt dependence reported by *Krainer et al., 2021* can be seen as an extension of the screening scenario, whereby high salt overcompensates the screening effect by strengthening π-type and hydrophobic interactions and leads to reemergence of phase separation. The high net charge scenario represented by A1-LCD is similar to the reentrant scenario at high salt but differs from it at low salt, where phase separation is prevented by net charge repulsion (*Martin et al., 2021*; *Muschol and Rosenberger, 1997*; *Kim et al., 2017*; *Dao et al., 2018*; *Babinchak et al., 2019*; *Le Ferrand et al., 2019*; *Wong et al., 2020*; *Agarwal et al., 2021*; *Otis and Sharpe, 2022*). In this class of salt dependence, a certain amount of salt is required to start phase separation. For A1-LCD, this minimum salt is ~100 mM NaCl (*Martin et al., 2021*). The recombinant mussel foot protein-1 (RMFP-1) presents an extreme example, which has a net charge of +24 over a sequence length of 121 and requires up to 700 mM NaCl to begin phase separation (*Kim et al., 2017*). The fourth class of salt dependence is a variation of the preceding one; here, the net charge is low and no significant net charge repulsion is expected, so the protein can phase separate even without salt. For example, FUS-LCD has a low net charge of −2 and phase separates without salt (*Burke et al., 2015*). The difference between the high and low net charge scenarios is well captured by the phase-separation behaviors of TDP43-LCD at two pHs (*Babinchak et al., 2019*). At pH 7, this protein has a relatively low net charge of +5 and phase separates without salt; the salt dependence thus belongs to the low net charge class. When pH is lowered to 4, the six His residues in the purification tag become positively charged, raising the net charge to +11; now phase separation requires 300 mM NaCl and the salt dependence switches to the high net charge class.

We can not only rationalize the four distinct classes of salt dependence but also start to predict them from the amino-acid composition of the protein (*Figure 6C* and *Table 1*). The foregoing physical understanding suggests three determinants of salt dependence class: (1) the total number of charged residues, which determines the contribution of electrostatic interactions to the drive for phase separation and the importance of salt screening; (2) the net charge, which determines the magnitude of net charge repulsion; and (3) the total number of aromatic residues, which determines whether phase separation reemerges at high salt. We predict that the screening class occurs when the charged content is high but the net charge and aromatic content are low (*Figure 6C*). The other three classes of salt dependence all require a high aromatic content. For the reentrant class, the increase in aromatic content is the only difference from the screening class. For the high net charge class, the net charge obviously has to be high, but that could also mean at least a moderately high charged content. Finally, the low net charge class is predicted when the net charge is low but no bias is required of the charged content. The amino-acid compositional data in *Table 1* validate these predictions. For example, of the 17 proteins in the classes of salt dependence that call for a high aromatic content, 12 (or 71%) actually have this feature (aromatic content >an 8% threshold). In contrast, of the 10 proteins in the class of salt dependence (i.e., 'screening') that calls for a low aromatic content, only 2 (or 20%) have an aromatic content above the threshold. In the latter two cases it remains to be tested whether phase separation would reemerge at much higher salt (and thereby resulting in a reclassification to reentrant). Likewise, of the 10 proteins in the class of salt dependence (i.e., 'high net charge') that calls for a high net charge, 7 (or 70%) actually have this feature (net charge >a 6% threshold). In contrast, of the 17 proteins in the classes of salt dependence that call for a low net charge, none has a net charge above

**Table 1.** Correlation between class of salt dependence and amino-acid composition.

| Protein | Length | Charges (+/−/net)* | Aromatic[†] | Salt (mM) | Ref |
|---|---|---|---|---|---|
| Screening | | | | | |
| FIB-1 | 352 | 55[‡]/**34**/21 | 24 | NaCl 50–250 | *Berry et al., 2015* |
| PolyQ | 729 | 38/34/4 | 35 | NaCl 0–150 | *Zhang et al., 2015* |
| Ddx4 | 236 | **32/36**/–4 | **22** | NaCl 100–500 | *Brady et al., 2017* |
| HP1α | 206 | **33/41**/–8 | 14 | NaCl 25–150 | *Strom et al., 2017* |
| LAF-1 | 708 | **86/88**/–2 | 55 | NaCl 125–400 | *Wei et al., 2017* |
| LAF-1 RGG | 191 | **28/22**/6 | 14 | NaCl 125–300 | *Wei et al., 2017* |
| Oleo30G | 139 | 15/12/3 | 7 | NaCl 35–280 | *Reed and Hammer, 2018* |
| FMRP-LCD | 188 | **37/28**[¶]/9 | 6 | NaCl 0–150 | *Tsang et al., 2019* |
| hnRNPA1 | 314 | **42/34**/8 | **33** | NaCl 50–300 | *Martin et al., 2021* |
| pY-Caprin1[§] | 103 | **16/17**/–1 | 4 | NaCl 100–500 | *Lin et al., 2024* |
| Reentrant | | | | | |
| FUS | 526 | 51/37/14 | **52** | KCl 50–2700 | *Krainer et al., 2021* |
| TDP-43 | 414 | **40/44**/–4 | 36 | KCl 50–2700 | *Krainer et al., 2021* |
| Brd4 | 1362 | **175/150**/25 | 48 | KCl 50–2150 | *Krainer et al., 2021* |
| Sox2 | 317 | 34/21/13 | 19 | KCl 50–2150 | *Krainer et al., 2021* |
| A11 | 505 | **51/50**/1 | **41** | NaCl 22.5–500 | *Krainer et al., 2021* |
| High net change | | | | | |
| A1-LCD | 131 | 12/3/**9** | **18** | NaCl 50–300 | *Martin et al., 2021* |
| Lysozyme[¶] | 129 | **17/9**/8 | **12** | NaCl 514–1198 | *Muschol and Rosenberger, 1997* |
| RMFP-1 | 121 | 24/0/**24** | **24** | NaCl 100–500 | *Kim et al., 2017* |
| UBQLN2 | 624 | 31/40/–9 | 22 | NaCl 100–300 | *Dao et al., 2018* |
| UBQLN2 (450-624) | 175 | 4/8/–4 | 5 | NaCl 50–200 | *Dao et al., 2018* |
| TDP43-LCD (pH 4) | 148 | 14[**]/3/**11** | **12** | NaCl 0–300 | *Babinchak et al., 2019* |
| HBP-2 (pH 5.5) | 193 | **30**[**]/**13/17** | **20** | NaCl 50–500 | *Le Ferrand et al., 2019* |
| Caprin1 | 103 | 16/3/**13** | **11** | NaCl 0–2000 | *Wong et al., 2020* |
| Prp-LCD | 122 | 11/1/**10** | **11** | NaCl 150–750 | *Agarwal et al., 2021* |
| RLP3₈ | 120 | 12/8/4 | 8 | NaCl 0–2000 | *Otis and Sharpe, 2022* |
| Low net charge | | | | | |
| FUS-LCD | 136 | 0/2/–2 | **20** | NaCl 0–250 | *Burke et al., 2015* |
| TDP43-LCD (pH 7) | 148 | 8/3 / 5 | **12** | NaCl 0–300 | *Babinchak et al., 2019* |

*Charges are listed as the number of positively (R and K, including H when specifically indicated) or negatively (D and E) charged residues or net charge.

[†]Aromatic residues are W, Y, and F.

[‡]Single-domain folded protein.

[§]Phosphorylated Y was assigned a charge of −2 and assumed to be no longer an aromatic residue.

[¶]High contents of charged residues, net charges, and aromatic residues are indicated by bold letters, with thresholds at 20%, 6%, and 8%, respectively, when measured as percentages of the sequence length.

[**]H is treated as positively charged at the low pH condition.

the threshold. Lastly, of the 15 proteins in the classes of salt dependence that call for a high charge content, 11 (or 73%) actually have this feature (charged content >a 20% threshold). In comparison, of the 12 proteins in the classes of salt dependence that do not require a high charge content, only 2 (or 17%) have a charge content above the threshold.

Two of the proteins included in *Table 1* are Caprin1 and its phosphorylated variant pY-Caprin1. Caprin1's phase separation required 500 mM NaCl and was promoted by higher salt (*Wong et al., 2020*), and thus behaved as if in the high net charge class of salt dependence (*Figure 6A*, third panel). In contrast, pY-Caprin1's phase separation was suppressed by salt (*Lin et al., 2024*), behaving as if in the screening class (*Figure 6A*, first panel). Our amino-acid composition-based predictor correctly places both proteins in these respective classes (*Table 1*), with Caprin1's sequence possessing both a high net charge and a high aromatic content whereas pY-Caprin1's sequence possessing a high level of charged residues. The disparate salt dependences of these two proteins were also correctly captured by both analytical theory and coarse-grained simulations (*Lin et al., 2024*). The simulations also showed the dominance of Cl$^-$ over Na$^+$ in partitioning into the dense phase of Caprin1, similar to our all-atom simulations of A1-LCD (*Figure 3B*); both Caprin1 and A1-LCD have a high net positive charge.

Comparing and contrasting coarse-grained simulations and all-atom simulations highlight the pros and cons of the two approaches. Coarse-grained simulations can readily capture the phase-separation process and represent the equilibrium between the dense and dilute phases. For short peptides, all-atom simulations have been shown to accomplish both of these tasks (*Zhang et al., 2024*). For the mixture of ATP and a 33-residue IDP, all-atom simulations were able to capture the condensation process but not the equilibration of the resulting dense phase with a dilute phase (*Kota et al., 2024*). In this work, although the total number of atoms in the A1-LCD system, ~half a million, is very high from a computational standpoint, the number of protein chains (i.e., 8) is still too low for a proper representation of a homogenous solution at low salt and a full equilibration between two phases at intermediate and high salt. Nevertheless, our simulations do not show condensation at low salt, thus mimicking a homogenous solution; the simulations at intermediate and high salt show condensation and thus model the dense phase. We are thus able to compare the salt effects in a homogenous solution (50 mM NaCl) and in the dense phase after phase separation (150–1000 mM NaCl) and compare the salt effects in dense phases with intermediate (150 and 300 mM NaCl) and high condensation (500 and 1000 mM NaCl). Importantly, the all-atom simulations are able to capture unusual physical effects that could be missed in coarse-grained simulations, such as strengthening π-types of interactions by drawing water away from the interaction partners. The latter effect is now confirmed by Raman spectroscopy (*Joshi et al., 2024*).

To conclude, salts regulate intermolecular interactions in biomolecular condensates in a variety of ways, but the net dependence of homotypic phase separation on salt concentration can be placed into four distinct classes (*Figure 6A, B*). Moreover, these classes are predictable from the protein amino-acid composition. Salts also exert significant effects on heterotypic condensates (*Farag et al., 2023*; *Galvanetto et al., 2023*); the conclusions drawn here on homotypic condensates may prove instructive for heterotypic condensates. The fact that different IDPs respond to salts differently raises interesting questions on phase separation inside cells. For example, a fluctuation in intracellular salt concentration may promote the phase separation of some IDPs and suppress the phase separation of other IDPs. If two IDPs co-phase separate or form heterotypic condensates, salt fluctuations may change the protein composition of condensates and the relative contributions of the individual proteins to phase separation. Lastly, we note that the atomistic details of ion coordination to A1-LCD backbone or sidechain groups are reminiscent of those observed in other systems, in particular channels and transporters for Na$^+$ and Cl$^-$ (*Mancusso et al., 2012*; *Mita et al., 2021*; *Leisle et al., 2022*).

## Methods
### MD simulations

MD simulations were performed using AMBER (*Case et al., 2018*) with ff14SB force field (*Maier et al., 2015*) for the protein and TIP4P-D for water (*Piana et al., 2015*). The initial configuration of the 8-chain system was constructed using A1-LCD conformations from previous single-copy simulations (*Dey et al., 2022*). Specifically, 4 copies with different conformations were placed in a rectangular

box, positioned with minimal inter-chain contacts within 3.5 Å. This 4-copy subsystem was duplicated to form the initial configuration of the 8-chain system (*Figure 1B*). The protein chains were solvated in a box with dimensions of 182 Å × 146 Å × 164 Å. For each desired salt concentration (50, 150, 300, 500, or 1000 mM), an appropriate number of water molecules were randomly selected and replaced with $Cl^-$ and $Na^+$ ions; excess $Cl^-$ was added to neutralize the system. The total number of atoms was ~500,000 for the 8-chain system at each salt concentration.

After energy minimization in sander (2000 steps of steepest descent and 3000 steps of conjugate gradient), each system was heated to 300 K over 100 ps with a 1 fs timestep, under constant NVT using the Langevin thermostat (*Pastor et al., 1988*) with a 3.0 $ps^{-1}$ damping constant. The simulation was then continued in four replicates at constant NPT for 1.5 μs with a 2 fs timestep. The final dimensions were stabilized at ~174 Å × 140 Å × 157 Å. Pressure was regulated using the Berendsen barostat (*Berendsen et al., 1984*) with a coupling constant of 2.0 ps. All simulations were run on GPUs using *pmemd.cuda* (*Salomon-Ferrer et al., 2013*). Bond lengths involving hydrogens were constrained using the SHAKE algorithm (*Ryckaert et al., 1977*). Long-range electrostatic interactions were treated by the particle mesh Ewald method (*Essmann et al., 1995*). A cutoff distance of 10 Å was used for the nonbonded interactions. Frames were saved every 200 ps and the last 500 ns was used for analysis.

## Data analysis

$D_{max}$ was calculated using the minmax function in VMD (*Humphrey et al., 1996*). The differences between the maximum and minimum values of $x$, $y$, and $z$ coordinates of all protein atoms were obtained; the largest of the three difference values was taken as $D_{max}$. To avoid issues with periodic boundary conditions, this calculation was repeated eight times, each time the simulation box was centered on one of the A1-LCD chains. The smallest of the 8 $D_{max}$ values is reported in *Figure 2A*. Radius of gyration ($R_g$) for each chain was calculated using the radgyr function in CPPTRAJ (*Roe and Cheatham, 2013*) and then averaged over the eight chains in each simulation (*Figure 2B*). For inter- and intrachain contacts per residue (*Figure 2C*), the number of other residues in contact with a given residue was found using the distance mask function in CPPTRAJ with a cutoff of 6 Å between heavy atoms. Again, to avoid issues with the periodic boundary conditions, each chain was centered individually before inter- and intrachain contacts were calculated for each residue within that chain. In the case of intrachain contacts, the two nearest neighboring residues of a given residue were excluded as contact partners. Inter-chain contacts per chain (*Figure 5A*) were found in a similar way using only sidechain heavy atoms, and were broken into interaction types based on the residue types of the interaction partners. Interaction types were defined as follows: salt bridges were between Arg/Lys and Asp, cation-π interactions were between Arg/Lys and Tyr/Phe, amino-π interactions were between Asn/Gln and Tyr/Phe, and π–π interactions were between a Tyr/Phe and another Tyr/Phe. To characterize chain reconfiguration, the RMSFs of the chains were calculated separately and then averaged over the 8 chains (*Figure 1—figure supplement 1*).

RDFs for ions around protein atoms were calculated in CPPTRAJ using the radial command with a bin size of 0.05 Å and a specific selection of the protein atom type for either $Cl^-$ or $Na^+$ at the center. These RDF plots were used to select cutoffs for 1st- and 2nd-shell ion binding; the cutoffs were 4.0 and 6.4 Å, respectively, for $Cl^-$ and 3.0 and 5.4 Å, respectively, for $Na^+$ (*Figure 3—figure supplements 1 and 2*). The number of 1st-shell or 1st- and 2nd-shell ions interacting with N and O atoms in a given sidechain type, the backbone, or the 8 chains together (*Figure 3*) was calculated using CPPTRAJ, by specifying a protein selection, an ion selection (all $Cl^-$ ions or all $Na^+$ ions), and a cutoff distance. Protein chains were centered one at a time and each time, the protein selection was limited to the central chain. The resulting ion–protein interactions for the eight chains were then combined and custom python scripts (https://github.com/hzhou43/Salt_on_A1LCD_LLPS, copy archived at *Zhou, 2024*) were used to collect unique ions in the total ion count for each protein selection (*Figure 3A*, right ordinate; *Figure 3B*). For ions interacting with a given sidechain type, the total ion count was normalized by the number of residues of that type to get the ion count per residue (*Figure 3A*, left ordinate). The net charge of the system was calculated by taking the total charge of the eight protein chains (+72) and adding and subtracting, respectively, the numbers of bound $Na^+$ and $Cl^-$ ions within the 2nd-shell cutoffs (*Figure 3C*). The total bound ions in *Figure 3B* were further divided according to the number of chains with which each ion was interacting in the same frame. Ions were considered

bridging if they interacted with two or more chains (based on 2nd-shell cutoffs) in the same frame (*Figure 4B*). Those interacting with an exact number of chains (e.g., 3) were identified when the same ion was found in the interaction lists of that many chains (*Figure 4C*). Error bars in all graphs represent the standard deviations among the four replicate simulations at each salt concentration.

RDFs for water molecules around Tyr residues were calculated using VMD with a bin size of 0.05 Å (*Figure 5E* and *Figure 5—figure supplement 1A–C*). The water atom selection was all its three atoms and the Tyr atom selection was its 6-carbon ring. Moreover, only Tyr residues that interact with a specific residue type (within 6 Å of the 6-carbon ring) were selected: the partner residues were Phe (6-carbon ring; for π–π), Arg and Lys (sidechain N atoms; for cation–π), or Gln and Asn (sidechain N atoms; for amino–π). As a control, RDFs for water were also calculated around Asp sidechain oxygens that interact with Arg and Lys sidechain nitrogens (*Figure 5—figure supplement 1D*).

## Acknowledgements

This work was supported by National Institutes of Health Grant GM118091.

## Additional information

### Funding

| Funder | Grant reference number | Author |
| --- | --- | --- |
| National Institute of General Medical Sciences | GM118091 | Huan-Xiang Zhou |

The funders had no role in study design, data collection, and interpretation, or the decision to submit the work for publication.

### Author contributions

Matt MacAinsh, Data curation, Formal analysis, Validation, Investigation, Visualization, Methodology, Writing - original draft; Souvik Dey, Conceptualization, Data curation, Investigation; Huan-Xiang Zhou, Conceptualization, Formal analysis, Supervision, Funding acquisition, Validation, Visualization, Writing - original draft, Project administration, Writing - review and editing

### Author ORCIDs

Huan-Xiang Zhou ⓘ https://orcid.org/0000-0001-9020-0302

Reviewer #1 (Public review): https://doi.org/10.7554/eLife.100282.3.sa1
Reviewer #2 (Public review): https://doi.org/10.7554/eLife.100282.3.sa2
Reviewer #3 (Public review): https://doi.org/10.7554/eLife.100282.3.sa3
Author response https://doi.org/10.7554/eLife.100282.3.sa4

## Additional files

### Supplementary files
• MDAR checklist

### Data availability
All data generated or analyzed during this study are included in the manuscript and supplementary files, including source data for figures.

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
