## [Editor Report · eLife Assessment]

In this potentially **important** study, the authors conducted atomistic simulations to probe the salt-dependent phase separation of the low-complexity domain of hnRN-PA1 (A1-LCD). The authors have identified both direct and indirect mechanisms of salt modulation, provided explanations for four distinct classes of salt dependence, and proposed a model for predicting protein properties from amino acid composition. There is a range of opinions regarding the strength of evidence, with some considering the evidence as **incomplete** due to the limitations in the length and statistical errors of the computationally intense atomistic MD simulations.

---

## [Referee Report · Reviewer #1 (Public review)]

Summary:

The authors examined the salt-dependent phase separation of the low-complexity domain of hnRN-PA1 (A1-LCD). Using all-atom molecular dynamics simulations, they identified four distinct classes of salt dependence in the phase separation of intrinsically disordered proteins (IDPs), which can be predicted based on their amino acid composition. However, the simulations and analysis, in their current form, are inadequate and incomplete.

Strengths:

The authors attempt to unravel the mechanistic insights into the interplay between salt and protein phase separation, which is important given the complex behavior of salt effects on this process. Their effort to correlate the influence of salt on the low-complexity domain of hnRNPA1 (A1-LCD) with a range of other proteins known to undergo salt-dependent phase separation is an interesting and valuable topic.

Weaknesses:

Based on the reviewer's assessment of the manuscript, the following points were raised:

(1) The simulation duration is too short to draw comprehensive conclusions about phase separation.

(2) There are concerns regarding the convergence of the simulations, particularly as highlighted in Figure 2A.

(3) The simulation begins with a protein concentration of 3.5 mM ("we built an 8-copy model for the dense phase (with an initial concentration of 3.5 mM)"), which is high for phase separation studies. The reviewer questions the use of the term "dense phase" and suggests that the authors conduct a clearer analysis depicting the coexistence of both the dilute and dense phases to represent a steady state. Without this, the realism of the described phenomena is doubtful. Commenting on phase separation under conditions that don't align with typical phase separation parameters is not acceptable.

(4) The inference that "Each Arg sidechain often coordinates two Cl- ions simultaneously, but each Lys sidechain coordinates only one Cl- ion" is questioned. According to Supplementary Figure 2A, Lys seems to coordinate with Cl- ions more frequently than Arg.

(5) The authors are requested to update the figure captions for Supplementary Figures 2 and 3, specifying which system the analyses were performed on.

(6) It is difficult to observe a clear trend due to irregularities in the data. Although the authors have included a red dotted line in the figures, the trend is not monotonic. The reviewer expresses concerns about significant conclusions drawn from these figures (e.g., Figure 2C, Figure 5A, Supplementary Figure 1).

(7) Given the error in the radius of gyration (Rg) calculations, the reviewer questions the validity of drawing conclusions from this data.

(8) The pair correlation function values in Figure 5E and supplementary figure 4 show only minor differences, and the reviewer questions whether these differences are significant.

(9) Previous reports suggest that, upon self-assembly, protein chains extend within the condensate, leading to a decrease in intramolecular contacts. However, the authors show an increase in intramolecular contacts with increasing salt concentration (Figure 2C), which contradicts prior studies. The reviewer advises the authors to carefully review this and provide justification.

(10) A systematic comparison of estimated parameters with varying salt concentrations is required. Additionally, the authors should provide potential differences in salt concentrations between the dilute and condensed phases.

(11) The reviewer finds that the majority of the data presented shows no significant alteration with changes in salt concentration, yet the authors have made strong conclusions regarding salt activity.

The manuscript lacks sufficient scientific details of the calculations.

---

## [Referee Report · Reviewer #2 (Public review)]

This is an interesting computational study addressing how salt affects the assembly of biomolecular condensates. The simulation data are valuable as they provide a degree of atomistic details regarding how small salt ions modulate interactions among intrinsically disordered proteins with charged residues, namely via Debye-like screening that weakens the effective electrostatic interactions among the polymers, or through bridging interactions that allow interactions between like charges from different polymer chains to become effectively attractive (as illustrated, e.g., by the radial distribution functions in Supplementary Information). However, this manuscript has several shortcomings: (i) Connotations of the manuscript notwithstanding, many of the authors' concepts about salt effects on biomolecular condensates have been put forth by theoretical models, at least back in 2020 and even earlier. Those earlier works afford extensive information such as considerations of salt concentrations inside and outside the condensate (tie-lines). But the authors do not appear to be aware of this body of prior works and therefore missed the opportunity to build on these previous advances and put the present work with its complementary advantages in structural details in the proper context. (ii) There are significant experimental findings regarding salt effects on condensate formation [which have been modeled more recently] that predate the A1-LCD system (ref.19) addressed by the present manuscript. This information should be included, e.g., in Table 1, for sound scholarship and completeness. (iii) The strengths and limitations of the authors' approach vis-à-vis other theoretical approaches should be discussed with some degree of thoroughness (e.g., how the smallness of the authors' simulation system may affect the nature of the "phase transition" and the information that can be gathered regarding salt concentration inside vs. outside the "condensate" etc.).

Comments on revised version:

The authors have adequately addressed my previous concerns and suggestions. The manuscript is now significantly improved. The new results and analyses provided by the authors represent a substantial advance in our understanding of the role of electrostatics in the assembly of biomolecular condensates.

---

## [Referee Report · Reviewer #3 (Public review)]

Summary:

This study investigates the salt-dependent phase separation of A1-LCD, an intrinsically disordered region of hnRNPA1 implicated in neurodegenerative diseases. The authors employ all-atom molecular dynamics (MD) simulations to elucidate the molecular mechanisms by which salt influences A1-LCD phase separation. Contrary to typical intrinsically disordered protein (IDP) behavior, A1-LCD phase separation is enhanced by NaCl concentrations above 100 mM. The authors identify two direct effects of salt: neutralization of the protein's net charge and bridging between protein chains, both promoting condensation. They also uncover an indirect effect, where high salt concentrations strengthen pi-type interactions by reducing water availability. These findings provide a detailed molecular picture of the complex interplay between electrostatic interactions, ion binding, and hydration in IDP phase separation.

Strengths:

• Novel Insight: The study challenges the prevailing view that salt generally suppresses IDP phase separation, highlighting A1-LCD's unique behavior.

• Rigorous Methodology: The authors utilize all-atom MD simulations, a powerful computational tool, to investigate the molecular details of salt-protein interactions.

• Comprehensive Analysis: The study systematically explores a wide range of salt concentrations, revealing a nuanced picture of salt effects on phase separation.

• Clear Presentation: The manuscript is well-written and logically structured, making the findings accessible to a broad audience.

Weaknesses:

• Limited Scope: The study focuses solely on the truncated A1-LCD, omitting simulations of the full-length protein. This limitation reduces the study's comparative value, as the authors note that the full-length protein exhibits typical salt-dependent behavior. However, given the much larger size of the full-length protein, it is acceptable to omit it given the current computing resources available.

Overall, this manuscript represents a significant contribution to the field of IDP phase separation. The authors' findings provide valuable insights into the molecular mechanisms by which salt modulates this process, with potential implications for understanding and treating neurodegenerative diseases.

---

## [Author Response]

The following is the authors’ response to the original reviews.

**Reviewer #1 (Public Review):**
Summary:The authors examined the salt-dependent phase separation of the low-complexity domain of hnRN-PA1 (A1-LCD). Using all-atom molecular dynamics simulations, they identified four distinct classes of salt dependence in the phase separation of intrinsically disordered proteins (IDPs), which can be predicted based on their amino acid composition. However, the simulations and analysis, in their current form, are inadequate and incomplete.Strengths:The authors attempt to unravel the mechanistic insights into the interplay between salt and protein phase separation, which is important given the complex behavior of salt effects on this process. Their effort to correlate the influence of salt on the low-complexity domain of hnRNPA1 (A1-LCD) with a range of other proteins known to undergo salt-dependent phase separation is an interesting and valuable topic.Weaknesses:(1) The simulations performed are not sufficiently long (Figure 2A) to accurately comment on phase separation behavior. The simulations do not appear to have converged well, indicating that the system has not reached a steady state, rendering the analysis of the trajectories unreliable.

We have extended the simulations for an additional 500 ns, to 1500 ns. The last 500 ns show reasonably good convergence (see Figure 2A).

(2) The majority of the data presented shows no significant alteration with changes in salt concentration. However, the authors have based conclusions and made significant comments regarding salt activities. The absence of error bars in the data representation raises questions about its reliability. Additionally, the manuscript lacks sufficient scientific details of the calculations.

We have now included error bars. With the error bars, the salt dependences of all the calculated properties (exception for Rg) show a clear trend. Additionally, we have expanded the descriptions of our calculations (p. 15-16).

(3) In Figures 2B and 2C, the changes in the radius of gyration and the number of contacts do not display significant variations with changes in salt concentration. The change in the radius of gyration with salt concentration is less than 1 Å, and the number of contacts does not change by at least 1. The authors' conclusions based on these minor changes seem unfounded.

The variation of ~ 1 Å for the calculated Rg is similar to the counterpart for the experimental Rg. As for the number of contacts, note that this property is presented on a per-residue basis, so a value of 1 means that each residue picks up one additional contact, or each protein chain gains a total of 131 contacts, when the salt concentration is increased from 50 to 1000 mM.

**Reviewer #2 (Public Review):**
This is an interesting computational study addressing how salt affects the assembly of biomolecular condensates. The simulation data are valuable as they provide a degree of atomistic details regarding how small salt ions modulate interactions among intrinsically disordered proteins with charged residues, namely via Debye-like screening that weakens the effective electrostatic interactions among the polymers, or through bridging interactions that allow interactions between like charges from different polymer chains to become effectively attractive (as illustrated, e.g., by the radial distribution functions in Supplementary Information). However, this manuscript has several shortcomings:(i) Connotations of the manuscript notwithstanding, many of the authors' concepts about salt effects on biomolecular condensates have been put forth by theoretical models, at least back in 2020 and even earlier. Those earlier works afford extensive information such as considerations of salt concentrations inside and outside the condensate (tie-lines). But the authors do not appear to be aware of this body of prior works and therefore missed the opportunity to build on these previous advances and put the present work with its complementary advantages in structural details in the proper context.(ii) There are significant experimental findings regarding salt effects on condensate formation [which have been modeled more recently] that predate the A1-LCD system (ref.19) addressed by the present manuscript. This information should be included, e.g., in Table 1, for sound scholarship and completeness.(iii) The strengths and limitations of the authors' approach vis-à-vis other theoretical approaches should be discussed with some degree of thoroughness (e.g., how the smallness of the authors' simulation system may affect the nature of the "phase transition" and the information that can be gathered regarding salt concentration inside vs. outside the "condensate" etc.). Accordingly, this manuscript should be revised to address the following. In particular, the discussion in the manuscript should be significantly expanded by including references mentioned below as well as other references pertinent to the issues raised.(1) The ability to use atomistic models to address the questions at hand is a strength of the present work. However, presumably because of the computational cost of such models, the "phase-separated" "condensates" in this manuscript are extremely small (only 8 chains). An inspection of Fig.1 indicates that while the high-salt configuration (snapshot, bottom right) is more compact and droplet-like than the low-salt configuration (top right), it is not clear that the 50 mM NaCl configuration can reasonably correspond to a dilute or homogeneous phase (without phase separation) or just a condensate with a lower protein concentration because the chains are still highly associated. One may argue that they become two droplets touching each other (the chains are not fully dispersed throughout the simulation box, unlike in typical coarse-grained simulations of biomolecular phase separation). While it may not be unfair to argue from this observation that the condensed phase is less stable at low salt, this raises critical questions about the adequacy of the approach as a stand-alone source of theoretical information. Accordingly, an informative discussion of the limitation of the authors' approach and comparisons with results from complementary approaches such as analytical theories and coarsegrained molecular dynamics will be instructive-even imperative, especially since such results exist in the literature (please see below).

We now discuss the limitations of our all-atom simulations and also other approaches (p. 13; see below).

(2) The aforementioned limitation is reflected by the authors' choice of using Dmax as a sort of phase separation order parameter. However, no evidence was shown to indicate that Dmax exhibits a twostate-like distribution expected of phase separation. It is also not clear whether a Dmax value corresponding to the linear dimension of the simulation box was ever encountered in the authors' simulated trajectories such that the chains can be reliably considered to be essentially fully dispersed as would be expected for the dilute phase. Moreover, as the authors have noted in the second paragraph of the Results, the variation of Dmax with simulation time does not show a monotonic rank order with salt concentration. The authors' explanation is equivalent to stipulating that the simulation system has not fully equilibrated, inevitably casting doubt on at least some of the conclusions drawn from the simulation data.

First off, with the extended simulations, the Dmax values converge to a tiered order rank, with successively decreasing values from low salt (50 mM) to intermediate salt (150 and 300 mM) to high salt (500 and 1000 mM). Secondly, as we now state (p. 13), our low-salt simulations mimic a homogenous solution whereas our high-salt simulations mimic the dense phase of a phase-separated system. The intermediate-salt simulations also mimic the dense phase but at a somewhat lower concentration (hence the intermediate Dmax value).

(3) With these limitations, is it realistic to estimate possible differences in salt concentration between the dilute and condensed phases in the present work? These features, including tie-lines, were shown to be amenable to analytical theory and coarse-grained molecular dynamics simulation (please see below).

The differences in salt effects that we report do not represent those between two phases. Rather, as explained in the preceding reply, they represent differences between a homogenous solution at low salt and the dense phase at higher salt. We also acknowledge salt effects calculated by analytical theory and coarse-grained simulations (p. 13).

(4) In the comparison in Fig.2B between experimental and simulated radius of gyration as a function of [NaCl], there is an outlier among the simulated radii of gyration at [NaCl] ~ 250 mM. An explanation should be offered.

After extending the simulations and analyzing the last 500 ns, the Rg data no longer show an outlier though still have some fluctuations from one salt concentration to another.

(5) The phenomenon of no phase separation at zero and low salt and phase separation at higher salt has been observed for the IDP Caprin1 and several of its mutants [Wong et al., J Am Chem Soc 142, 24712489 (2020) [https://pubs.acs.org/doi/full/10.1021/jacs.9b12208], see especially Fig.9 of this reference]. This work should be included in the discussion and added to Table 1.

We now have added Caprin1 to Table 1 (new ref 26) and discuss this paper (p. 13).

(6) The authors stated in the Introduction that "A unifying understanding of how salt affects the phase separation of IDPs is still lacking". While it is definitely true that much remains to be learned about salt effects on IDP phase separation, the advances that have already been made regarding salt effects on IDP phase separation is more abundant than that conveyed by this narrative. For instance, an analytical theory termed rG-RPA was put forth in 2020 to provide a uniform (unified) treatment of salt, pH, and sequence-charge-pattern effects on polyampholytes and polyelectrolytes (corresponding to the authors' low net charge and high net charge cases). This theory offers a means to predict salt-IDP tie-lines and a comprehensive account of salt effect on polyelectrolytes resulting in a lack of phase separation at extremely low salt and subsequent salt-enhanced phase separation (similar to the case the authors studied here) and in some cases re-entrant phase separation or dissolution [Lin et al., J Chem Phys 152. 045102 (2020) [https://doi.org/10.1063/1.5139661]]. This work is highly relevant and it already provided a conceptual framework for the authors' atomistic results and subsequent discussion. As such, it should definitely be a part of the authors' discussion.

We now cite this paper (new ref 34) in Introduction (p. 4). We also discuss its results for Caprin1 (new ref 18; p. 13).

(7) Bridging interactions by small ions resulting in effective attractive interactions among polyelectrolytes leading to their phase separation have been demonstrated computationally by Orkoulas et al., Phys Rev Lett 90, 048303 (2003) [https://journals.aps.org/prl/abstract/10.1103/PhysRevLett.90.048303]. This result should also be included in the discussion.

We now cite this paper (new ref 41; p. 11).

(8) More recently, the salt-dependent phase separations of Caprin1, its RtoK variants and phosphorylated variant (see item #5 above) were modeled (and rationalized) quite comprehensively using rG-RPA, field-theoretic simulation, and coarse-grained molecular dynamics [Lin et al., arXiv:2401.04873 [https://arxiv.org/abs/2401.04873]], providing additional data supporting a conceptual perspective put forth in Lin et al. J Chem Phys 2020 (e.g., salt-IDP tie-lines, bridging interactions, reentrance behaviors etc.) as well as in the authors' current manuscript. It will be very helpful to the readers of eLife to include this preprint in the authors' discussion, perhaps as per the authors' discretion along the manner in which other preprints are referenced and discussed in the current version of the manuscript.

We now cite this paper (new ref 18) and discuss it along with new ref 26 in Discussion (p. 13).

**Reviewer #3 (Public Review):**
Summary:This study investigates the salt-dependent phase separation of A1-LCD, an intrinsically disordered region of hnRNPA1 implicated in neurodegenerative diseases. The authors employ all-atom molecular dynamics (MD) simulations to elucidate the molecular mechanisms by which salt influences A1-LCD phase separation. Contrary to typical intrinsically disordered protein (IDP) behavior, A1-LCD phase separation is enhanced by NaCl concentrations above 100 mM. The authors identify two direct effects of salt: neutralization of the protein's net charge and bridging between protein chains, both promoting condensation. They also uncover an indirect effect, where high salt concentrations strengthen pi-type interactions by reducing water availability. These findings provide a detailed molecular picture of the complex interplay between electrostatic interactions, ion binding, and hydration in IDP phase separation.Strengths:Novel Insight: The study challenges the prevailing view that salt generally suppresses IDP phase separation, highlighting A1-LCD's unique behavior.Rigorous Methodology: The authors utilize all-atom MD simulations, a powerful computational tool, to investigate the molecular details of salt-protein interactions.Comprehensive Analysis: The study systematically explores a wide range of salt concentrations, revealing a nuanced picture of salt effects on phase separation.Clear Presentation: The manuscript is well-written and logically structured, making the findings accessible to a broad audience.Weaknesses:Limited Scope: The study focuses solely on the truncated A1-LCD, omitting simulations of the full-length protein. This limitation reduces the study's comparative value, as the authors note that the full-length protein exhibits typical salt-dependent behavior. A comparative analysis would strengthen the manuscript's conclusions and broaden its impact.

Perhaps we did not impress on the reviewer how expensive the all-atom MD simulations on A1-LCD were: the systems each contained half a million atoms and the simulations took many months to complete. That said, we agree with the reviewer that, ideally, a comparative study on a protein showing the typical screening class of salt dependence would have made our work more complete. However, we are confident of the conclusions for several reasons. First, the three salt effects – charge neutralization, bridging, and strengthening of pi-types of interactions – revealed by the all-atom simulations are physically sound and well-supported by other studies. Second, these effects led us to develop a unified picture for the salt dependence of homotypic phase separation, in the form of a predictor for the classes of salt dependence based on amino-acid composition. This predictor works well for nearly 30 proteins. Third, recent studies using analytical theory and coarse-grained simulations (new ref 18) also strongly support our conclusions.

**Reviewer #1 (Recommendations For The Authors):**
(1) In Figure 1, the color scheme should be updated and the figure remade, as the current set of color choices makes it very difficult to distinguish the magenta spheres.

We have increased the sizes of ions in Figure 1 to make them distinguishable.

(2) Within the framework of atomistic simulations, the influence of salt concentration alteration on protein conformational plasticity is worth investigating. This could be correlated (with proper details) with the effect of salt-concentration-modulated protein aggregation behavior.

We now use RMSF to measure conformational plasticity, which shows a clear salt-dependent trend with a 27% reduction in fluctuations from 50 mM to 1000 mM NaCl (new Fig. S1).

(3) The authors should mention the protein concentrations employed in the simulations and whether these are consistent with experimentally used concentrations.

We have mentioned the initial concentration (3.5 mM). We now further state that this concentration is maintained in the low-salt simulations, indicating absence of phase separation, but is increased to 23 mM in the high-salt simulations, indicating phase separation. The latter value is consistent with the measured concentrations in the dense phase (last two paragraphs of p. 5).

(4) It would be useful to test the salt effect for at least two extreme salt concentrations at various protein concentrations, consistent with experimental protein concentration ranges.

In simulation studies of short peptides (ref 37), we have shown that the initial concentration does not affect the final concentration in the dense phase, as expected for phase-separation systems. We expect that the same will be true for the A1-LCD system at intermediate and high salt where phase separation occurs. Though this expectation could be tested by simulations at a different initial protein concentration, such simulations would be expensive but unlikely to yield new physical insight.

(5) Importantly, the simulations do not appear to have converged well enough (Figure 2A). The authors should extend the simulation trajectories to ensure the system has reached a steady state.

We extended the simulations for an additional 500 ns, which now appear to show convergence. In Figure 2A we now see Dmax values converge to a tiered order rank, with successively decreasing values from low salt (50 mM) to intermediate salt (150 and 300 mM) to high salt (500 and 1000 mM).

(6) The authors mention "phase separation" in the title, but with only a 1 μs simulation trajectory, it is not possible to simulate a phenomenon like phase separation accurately. Since atomistic simulations cannot realistically capture phase separation on this timescale, a coarse-grained approach is more suitable. To properly explore salt effects in the context of phase separation, long timescale simulation trajectories should be considered. Otherwise, the data remain unreliable.

Our all-atom simulations revealed rich salt effects that might have been missed in coarse-grained simulations. It is true that coarse-grained models allow the simulations of the phase separation process, but as we have recently demonstrated (refs 36 and 37), all-atom simulations on the μs timescale are also able to capture the spontaneous phase separation of peptides and small IDPs. A1-LCD is much larger than those systems, so we had to use a relatively small chain number (8 chains here vs 64 used in ref 37 and 16 used in ref 37). S2ll, we observe the condensation into a dense phase at high salt. We discuss the pros and cons of all-atom vs. coarse-grained simulations in p. 13.

(7) In Figure 5E, the plot does not show that g(r) has reached 1. If it does, the authors should show the full curve. The same issue remains with supplementary figures 1, 2, 3, etc.

We now show the approach to 1 in the insets of Figs. S2, S3, S4, and 5E.

(8) None of the data is represented with error bars. The authors should include error bars in their data representations.

We have now included error bars in all graphs that report average values.

(9) The authors state that "the net charge of the system reduces to only +8 at 1000 mM NaCl (Figure 3C)" but do not explain how this was calculated.

We now add this explanation in methods (p. 16).

(10). The authors mention "similar to the role played by ATP molecules in driving phase separation of positively charged IDPs." However, ATP can inhibit aggregation, and its induction of phase separation is concentration-dependent. Given ATP's large aromatic moiety, its comparison to ions is not straightforward and is more complex. This comparison can be at best avoided.

In this context we are comparing the bridging capability of ATP molecules in driving phase separation of positively charged IDPs in ref 36 to the bridging capability of the ions here. In ref 36 the authors show ATP bridging interactions between protein chains similar to what we show here with ions.

(11) Many calculations are vaguely represented. The process for calculating the number of bridging ions, for example, is not well documented. The authors should provide sufficient details to allow for the reproducibility of the data.

We have now expanded the methods section to include more detailed information on calculations done.

**Reviewer #3 (Recommendations For The Authors):**
Include error bars or standard deviations for all results averaged over four replicates, particularly for the number of ions and contacts per residue. This would provide a clearer picture of the data's reliability and variability.

We have now included error bars in all graphs that report averaged values.

Strengthen the support for the conclusion that "each Arg sidechain often coordinates two Cl- ions, multiple backbone carbonyls often coordinate a single Na+ ion." While Fig. 3A clearly demonstrates ArgCl- coordination, the Na+ coordination claim for a 131-residue protein requires further clarification. Consider including the integration profile of radial distribution functions for Na+ ions to bolster this assertion.

We now report the number of Na+ ions that coordinate with multiple backbone carbonyls (p. 7) as well as the number of Na+ ions that bridge between A1-LCD chains via coordination with multiple backbone carbonyls (p. 9). Please note that Figure 4A right panel displays an example of Na+ coordinating with multiple backbone carbonyls.

Address the following typographical errors in the main text: o Page 11, line 25: "distinct classes of sat dependence" should be "distinct classes of salt dependence" o Page 14, line 9: "for Cl- and 3.0 and 5.4 A" should be "for Cl- and 3.0 and 5.4 √Ö" o Page 14, line 18: "As a control, PRDFs for water were also calculated" should be "As a control, RDFs for water were also calculated" (assuming PRDF was meant to be RDF)

We have now corrected these typos.

Consider expanding the study to include simulations of the full-length protein to provide a more comprehensive comparison between the truncated A1-LCD and the complete protein's behavior in various salt concentrations.

As we explained above, even with eight chains of A1-LCD, which has 131 residues, the systems already contain half a million atoms each and the all-atom simulations took many months to complete. Full-length A1 has 314 residues so a multi-chain system would be too large to be feasible for all-atom simulations.